# The Role of Diet and Nutrition in Allergic Diseases

**DOI:** 10.3390/nu15173683

**Published:** 2023-08-22

**Authors:** Ping Zhang

**Affiliations:** Center for Integrative Conservation, Yunnan Key Laboratory for the Conservation of Tropical Rainforests and Asian Elephants, Xishuangbanna Tropical Botanical Garden, Chinese Academy of Sciences, Xishuangbanna 6663030, China; zhangping@xtbg.org.cn

**Keywords:** allergy, allergic inflammation, asthma, allergic rhinitis, atopic dermatitis, dietary lipids, dietary fiber, dietary flavonoids, micronutrients

## Abstract

Allergic diseases are a set of chronic inflammatory disorders of lung, skin, and nose epithelium characterized by aberrant IgE and Th2 cytokine-mediated immune responses to exposed allergens. The prevalence of allergic diseases, including asthma, allergic rhinitis, and atopic dermatitis, has increased dramatically worldwide in the past several decades. Evidence suggests that diet and nutrition play a key role in the development and severity of allergic diseases. Dietary components can differentially regulate allergic inflammation pathways through host and gut microbiota-derived metabolites, therefore influencing allergy outcomes in positive or negative ways. A broad range of nutrients and dietary components (vitamins A, D, and E, minerals Zn, Iron, and Se, dietary fiber, fatty acids, and phytochemicals) are found to be effective in the prevention or treatment of allergic diseases through the suppression of type 2 inflammation. This paper aims to review recent advances in the role of diet and nutrition in the etiology of allergies, nutritional regulation of allergic inflammation, and clinical findings about nutrient supplementation in treating allergic diseases. The current literature suggests the potential efficacy of plant-based diets in reducing allergic symptoms. Further clinical trials are warranted to examine the potential beneficial effects of plant-based diets and anti-allergic nutrients in the prevention and management of allergic diseases.

## 1. Introduction

Allergic diseases are a set of disorders caused by aberrant IgE-mediated immune responses to exposed allergens, resulting in clinical symptoms such as red itchy eyes, sneezing, nasal congestion, rhinorrhea, coughing, and itchy swollen skin [1]. The prevalence of allergic diseases, including asthma, allergic rhinitis (AR), and atopic dermatitis (AD), is high in developed countries [2,3,4], and the dramatically increased incidence of allergic diseases in developing countries may be due to a shift in lifestyle towards Western customs [5,6]. In allergic diseases, a complex interaction between genetic and environmental factors leads to abnormal immune responses at barrier sites in the body [2,3,4]. The Western diet is recognized as an environmental risk factor for developing allergic diseases [4,5,6], whereas the Mediterranean diet has been found to be protective [5,7,8]. Therefore, due to the opposite effects in allergic reactions conferred by different dietary components, diets with different nutrient compositions and varied amounts of specific nutrients either promote sensitization and exacerbate disease severity or protect against allergic diseases and attenuate disease progression. There has been growing interest in dissecting the connection between nutrients, their metabolites, and immune tolerance in allergic conditions. 

Apart from diet and nutrition, gut microbiota has recently been linked with allergic diseases [9,10]. Diet and food components play critical roles in shaping the gut microbiota, which is essential in maintaining the integrity of the gut epithelial barrier and gut immune homeostasis [11,12]. Moreover, nutrients and their endogenous or bacterial metabolites can regulate allergic inflammation in distant organs beyond the gut, such as the lung and skin through the gut–lung and gut–skin axes [13,14]. Among bacterial metabolites, short-chain fatty acids (SCFAs), bile acid conjugates, and tryptophan metabolites are the most studied compounds with the ability to modify allergic reactions [8,13,14]. Multiple cells including epithelial cells, stromal cells, sensory nerve cells, and various immune cells are involved in a typical allergic reaction with a signature Th2 cytokine profile and allergic inflammatory mediators including histamines, prostaglandins, and leukotrienes [2,3,4]. Nutrients and their metabolites can regulate the metabolism and function of both structural cells and various immune cells in all stages of allergic inflammation by altering the membrane lipid composition, key signal transduction pathways related to inflammation and metabolism, and gene expression at the transcriptional level through epigenetic regulation. The impacts of dietary components on allergic reactions are illustrated in Figure 1.

Accumulating evidence has shown that a broad range of nutrients and dietary components (vitamins A, D, and E, minerals Zn and iron, dietary fiber, fatty acids, and phytochemicals) play critical roles in the prevention or treatment of allergic disease through host and gut microbiota-derived metabolites. The purpose of this paper is to review recent advances in the understanding of diet and food components as contributing factors in the etiology of allergies, molecular targets of nutrient regulation of immune cells and structural cells involved in allergy, and clinical findings about nutrition intervention in treating allergic diseases.

## 2. Materials and Methods

A systematic literature search was conducted for reports in English from January 2013 to August 2023 using PubMed and Web of Science databases. The following key words were used individually or in combination: allergy, asthma, allergic rhinitis, atopic dermatitis, dietary fiber, dietary lipids, dietary protein, dietary flavonoids, micronutrients, obesity, and plant-based diet. Relevant articles were reviewed, and the most recent ones were preferably cited. Additional reports were identified from selected papers in the reference list. In general, priority was given to original research and review articles based on animal studies and clinical trials. 

## 3. Pathophysiology of Allergic Diseases

All allergic diseases involve type 2 inflammatory allergic responses to various allergens. The prototypical allergic reaction includes a sensitization and memory phase and an effector phase [15]. Common environmental allergens include dust mites, fungi, pets, and pollens [3]. During the sensitization phase, allergens entering through the epithelial barrier, where damage is caused by viruses or other environmental factors, are captured by dendritic cells and presented to naïve CD4^+^ T cells, leading to the generation of allergen-specific CD4^+^ Th2 cells which produce IL-4, IL-5, IL-9, and IL-13 [3,15]. Epithelial cells sense the danger and release three cytokines, TSLP, IL-33, and IL-25, which create a cytokine milieu to promote the generation of Th2 cells [16]. Besides epithelial cells, stromal cells can also sense changes in metabolite levels and secrete IL-33 in response to abnormal metabolite profiles [13,17]. High-level IL-4 and IL-13 induce IgE isotype class-switching in B cells, which will produce large amounts of IgE when matured into antigen-specific plasma cells. IgE binds through high-affinity FcεRI receptors on the surface of specific innate effector cells (mast cells and basophils). At this stage, a memory pool of antigen-specific Th2 cells and B cells is generated [3,15]. During the acute effector phase, an encounter with the allergen induces the cross-linking of the IgE on the surface of sensitized effector cells, triggering activation of effector cells and the release of mediators including preformed histamine and tryptase, and de novo synthesized prostaglandin D_2_ (PGD_2_) and leukotrienes C_4_ (LTC_4_), LTD_4_, and LTE_4_ [2,3]. These mediators interact with sensory nerve cells, glandular cells, and epithelial cells to generate acute symptoms such as itching, sneezing, coughing, and diarrhea in mucosal tissues [3]. In the later effector phase, accumulation of the above mediators released by innate immune cells, together with cytokines IL-4, IL-5, IL-9, and IL-13 produced by Th2 cells and type 2 innate lymphoid cells (ILC2s), as well as epithelial cell-derived cytokines, maintain high antigen-specific IgE levels and recruit more inflammatory cells including eosinophils and basophils into inflamed tissue, resulting in tissue damage and chronic inflammation in a type I hypersensitivity reaction. 

Epithelial cell-derived TSLP, IL-33, and IL-25 are critical initiators of type 2 immunity; however, their function is beyond merely sending an alarm signal [16]. They regulate a broad range of immune cells including the activation of dendritic cells to present antigens to naïve T cells, promoting Th2 cell development, stimulating neuron cells, activating ILCs, and enhancing memory Th2 cells [16]. Therefore, targeting these alarmins may be effective in lowering susceptibility and decreasing exacerbations in all allergic conditions. In fact, diet can influence the production of alarmins. For example, a high-fat diet promotes serum TSLP [18] and a high inulin fiber diet upregulates IL-33 from stromal cells through gut microbiota-derived bile acids [13]. In contrast, dietary fish oil or fermented fish oil (both are enriched with long-chain unsaturated fatty acids EPA (eicosapentaenoic acid) and DHA (docosahexaenoic)) lowers TSLP expression in mouse ear tissue with AD [19], and a natural flavonoid quercetin lowers TSLP levels in an in vitro AD model using human keratinocytes [20]. 

Innate lymphoid cells (ILCs) are tissue-resident innate immune cells that regulate tissue-specific immunity through interactions with epithelial cells, neurons, stromal cells, and other tissue-resident cells [21]. ILC2 cells are highly enriched in mucosal sites such as the lung, skin, and gut and are essential in type 2 inflammation. They are rapidly activated by TSLP, IL-33, and IL25 and produce high levels of the classical Th2 cytokines IL-4, IL-5, IL-9, and IL-13, therefore driving the pathogenesis of allergic diseases such as asthma, AR, and AD. Some dietary metabolites, such as retinoic acid in carrots and indole-3-carbinol contained in cabbage and broccoli [22,23], can restrain ILC2 responses through the activation of the aryl hydrocarbon receptor (AhR). The benefits of consuming these vegetables in the prevention of allergic diseases are likely due to these AhR ligands. Dietary factors can affect ILC2 cells through other mechanisms besides acting as AhR ligands. For example, dietary fiber metabolite butyrate can inhibit ILC2 proliferation and inhibit IL-13 and IL-5 production from ILC2 cells through histone deacetylase (HDAC) inhibition. Therefore, systemic administration of butyrate through drinking water or intranasal administration can attenuate ILC2-driven airway inflammation and airway hypersensitivity [24]. 

Allergen-specific regulatory T cells (Tregs) and regulatory B cells (Bregs) play essential roles in the induction of immune tolerance to allergens and restoring immune homeostasis in allergen-specific immunotherapy [15]. CD4^+^FOXP3^+^CD25^+^ Tregs can suppress ongoing allergic inflammation by inhibiting DCs, effector Th (Th1, Th2, and Th17) cells, granulocytes (mast cells, basophils, and eosinophils), B cells, as well as tissue-resident cells, either through secreted inhibitory cytokines (IL-10, TGF-β) or through cell contact-dependent mechanisms [15]. Bregs also play a key role in maintaining tolerance to allergens through the production of anti-inflammatory IgG4 antibodies and by secretion of suppressive cytokines IL-10, TGF-β, and IL-35 which promote Treg generation, inhibit T cell activation, and induce tolerogenic DCs [15]. Nutrient metabolism can influence Treg or Breg generation and function. For example, indoleamine 2, 3-dioxygenase (IDO), a key enzyme responsible for catabolizing dietary tryptophan to kynurenines, is highly expressed in dendritic cells in nose-draining lymph nodes and is essential to immune tolerance of inhaled allergens. A blockade of IDO impairs Treg differentiation during intranasal allergen challenge, which leads to the abrogation of allergen-specific immune tolerance [25]. A lower IDO level is associated with atopy in humans [26]. Moreover, maternal tryptophan metabolism can influence the development of allergic diseases in offspring [27]. Decreased numbers of regulatory B cells or functional changes in them are also observed in patients with allergic disorders including AR, asthma, and AD [28,29,30]. In patients with AR, decreased IL-10-secreting Bregs are linked to altered glutamine metabolism [31]. Both retinoic acid metabolized from vitamin A [32] and 1, 25-dihyroxyvitamin D_3_ metabolized from vitamin D_3_ [33] promote Foxp3^+^ Treg differentiation and immune suppression of T helper cells. Deficiency of dietary vitamin A or vitamin D induces high levels of Th2 cytokines and IgE responses to allergens [34,35]. Fermented fish oil suppresses allergic inflammation in the skin, at least partly through enhancing TGF-β and IL-10 expression, which might lead to tissue-specific Foxp3^+^ Tregs [19]. The trace mineral Zn also promotes Treg differentiation [36,37] and therefore is essential to immune tolerance of allergens. AhR is highly expressed on various antigen-presenting cells [38,39], and activation of AhR has been shown to promote Treg generation through induction of tolerogenic DC [38,40] or promote IL-10-producing Breg differentiation and function [41]. Recent studies in mice showed dietary supplements of whey-protein-derived β-lactoglobulin complexed with quercetin-iron or catechine-iron to be effective for reducing allergic symptoms [42,43]. Activation of AhR by quercetin or catechine, along with increased Tregs, are associated with the observed beneficial effects [42,43]. 

Allergic rhinitis (AR) is an inflammation of the nasal mucosa associated with an IgE-mediated response to environmental allergens and characterized by nasal itching, sneezing, rhinorrhea, and nasal congestion. AR is often co-morbid with asthma and conjunctivitis [3]. It is one of the most common chronic inflammatory conditions and a global health problem affecting over 500 million people worldwide [44]. In Europe, the prevalence of AR in some European countries can be as high as 50% of the population [3]. In China, the prevalence of AR ranged from 6.2% to 7.2% in adults living in rural and urban areas, respectively, in 2015 [45]. In Taiwan, the prevalence of AR was much higher, with 28.6% and 19.5% in men and women in 1995 [6]. A higher average income in Taiwan, as opposed to mainland China, could be a contributing factor. According to a recent survey in the city of Urugaiana, southern Brazil, the prevalence of AR was 31.7% in adults and 28% in adolescents [46]. Although not life-threatening, AR impairs the patient’s quality of life, lowers work performance and sleep quality, and therefore can result in substantial economic costs [3]. In AR, initial allergen exposure leads to damage in the nasal epithelial cells and the generation of allergen-specific IgE antibodies and Th2 memory cells. Upon re-exposure to the allergen, crosslinking of IgE on mast cells and basophils results in degranulation and the release of mediators of hypersensitivity which produce immediate nasal symptoms within minutes [47]. The late-phase nasal symptoms, such as nasal blockage and nasal discharge, happen within hours and are mainly caused by recruited eosinophils [47]. CD4^+^ Th2 cells, B cells, mast cells, neutrophils, and macrophages are observed in the nasal lining infiltrate [47]. Many epidemiological and clinical studies supported the role of diet and nutrition in the etiology, prevention, and treatment of AR [6,46,48,49,50,51].

Allergic asthma is the most common inflammatory disease of the lungs, with respiratory symptoms such as wheezing, shortness of breath, chest tightness and coughing, and airway hyper-responsiveness to inhaled allergens [2]. The prevalence of asthma in Western countries plateaued at 10% in recent decades. In contrast, the prevalence of asthma in countries with low and medium gross domestic product (GDP) has had a sharp increase in recent years [2] in contrast to previously much lower incidence statistics, making asthma a worldwide inflammatory disease. With eosinophils as the main airway infiltrate cell type, other cells including mast cells, basophils, neutrophils, monocytes, and macrophages can also be found [2]. Apart from airway inflammation, airway remodeling is another feature of asthma that involves structural changes such as subepithelial basement membrane thickening, subepithelial fibrosis, goblet cell hyperplasia and hypertrophy, and muscle hyperplasia [2]. Airway remodeling parallels disease development and leads to lung function decline. None of the current drug therapies can alter the natural history of asthma [2]. The impact of diet on asthma has been described [5] and most studies in the past were focused on the relationship between nutrients and airway inflammation. However, recent studies show evidence that dietary phytochemicals such as resveratrol [52] and kaempferol [53] can modify airway inflammation as well as airway remodeling, suggesting potential therapeutic value in treating allergic asthma. Some in vitro studies also showed that vitamin D is likely to play a role in airway remodeling in asthma [54].

Atopic dermatitis (AD) is a chronic inflammatory skin disease characterized by intense itching and eczematous lesions. Although recognized as an early onset disease as the first step of the so-called atopic march, it can start later in life and is quite common in adults [4]. It is one of the most common chronic inflammatory diseases, affecting 10–20% of the population in developed countries and its prevalence in developing countries continues to rise [4]. Although originally thought to be a typical allergic disorder, skin barrier dysfunction is discovered to be a key driver of AD [4,55,56]. Current research emphasis shifts from focusing on immune mechanisms to epidermal barrier dysfunction. Abnormal skin structure and altered lipid composition, inherited filaggrin deficiency, and environmental factors such as detergent use and mite allergens all contribute to skin barrier dysfunction in AD [4,55,56]. Skin infiltration of inflammatory cells mainly consists of Th2, Th22, and Th17 cells, together with ILC2 cells [4]. Nutrition plays critical roles in the etiology, prevention, and treatment of AD [57]. For example, a high-fat diet exacerbates AD through upregulation of TSLP [18]. A sufficient level of Vitamin D is essential for the maintenance of a normal skin barrier and vitamin D supplements are considered an alternative strategy for controlling skin barrier dysfunction in AD and the atopic march [55]. The role of dietary fiber in the prevention of AD recently emerged from a preclinical study in a mouse model of AD [14]. A high-fiber diet, or a low-fiber diet with orally administered SCFAs, protected against allergen-induced skin inflammation and allergen sensitization [14]. The underlying mechanism lies in gut-derived SCFAs, particularly butyrate, which promote skin barrier integrity by modulating keratinocyte metabolism and differentiation [14].

## 4. The Role of Diet and Nutritional Status in Allergy

Dietary factors not only affect the development of allergic diseases [5,6,46,50] but also influence disease course and severity [50,58]. Different dietary components are related to differential allergy outcomes. The intake of high energy, high saturated fat, high protein, and low fiber increases the risks of asthma and AR [6,46]. In contrast, high consumption of vegetables and fruits, olive oil, and fish, characteristic of a Mediterranean diet, has been linked with lower risks of asthma and AR [5,7,8,46,59]. Recent evidence suggests that higher dietary fiber intake is associated with fewer asthma symptoms [58]. Moreover, adequate intake of micronutrients is associated with a lower risk of atopic diseases and reduction of symptoms [50]. The identified diet and nutritional risk factors for allergy are shown in Box 1.

Box 1Diet and Nutritional Risk Factors for Allergy.High energyHigh proteinHigh saturated fat, *n*-6 fatty acids, medium-chain fatty acids, cholesterolLow total dietary fiberLow vegetables and fruitsHigh simple sugar and processed foodsLow level of Zn, Fe, Vitamins A, D, E

There is a close connection between nutrient metabolism and allergic diseases. Broad changes in energy, amino acids, and lipid metabolism are found in patients with pollinosis [60]. Patients with AR are shown to have at least 10 elevated metabolites in serum which belong to three pathways, namely, porphyrin and chlorophyll, arachidonic acid, and purine metabolism [61]. More and more cellular and molecular mechanisms are being elucidated concerning the regulation of allergic inflammation by individual dietary components or specific nutrients (Figure 2). The pro-allergic nutrients, such as saturated fatty acids and cholesterol, promote the release of TSLP, IL-25, and IL-33 from epithelial and stromal cells, and activate ILC2 cells to produce IL-4, IL-5, IL-9, and IL-13, therefore producing a cytokine milieu for allergic inflammation. By contrast, anti-allergic nutrients, including phytochemicals, micronutrients, and dietary fiber, can suppress allergic inflammation through inhibition of type 2 cytokine production in ILC2 cells via activation of AhR, promotion of the generation of tolerogenic dendritic cells, anti-inflammatory macrophages, and Tregs, and suppression of the release of histamine, prostaglandins, and leukotrienes from granulocytes.

### 4.1. Dietary protein, Amino Acids, and Energy

A high-protein diet is associated with an increased risk for type 1 allergy in OVA-sensitized mice, as indicated by increased B cells, total and antigen-specific IgE, and a skewed Th1/Th2 balance towards Th2 dominance [62]. In these mice, moderate protein deficiency without energy restriction results in similar total IgE as a normal protein diet [62], suggesting that energy is critical in regulating IgE production and limiting energy supply is important in controlling high IgE response during the exacerbation period in allergic diseases. Indeed, 40% dietary energy restriction delayed the onset of spontaneous dermatitis in NC/Nga (Nagogy University mice) mice whichs resemble human AD [63]. Moreover, dietary restriction suppressed the progression of dermatitis in these mice and was associated with reduced serum IgE, with much fewer numbers of infiltrating inflammatory cells (lymphocytes and eosinophils) in the skin and decreased dermal IL-4 and IL-5 production [63]. The effects of energy or protein restriction on other allergic diseases remain to be investigated.

The essential amino acid tryptophan is a key regulator of immune tolerance. Tryptophan is metabolized to kynurenine by IDO (indolamin 2, 3-dioxygenase) in DCs and binds to AhR on naïve CD4^+^ T cells to generate FoxP3^+^ Treg cells [25]. Expression of IDO is much higher in nose-draining lymph nodes, i.e., cervical lymph nodes, compared with peripheral lymph nodes [25]. In a mouse model of OVA-induced delayed hypersensitivity, inhibition of IDO during intranasal OVA administration results in the loss of immune tolerance as indicated by the increase in ear thickness [25]. IDO blockade was associated with dysfunctional Tregs which failed to suppress DTH (delayed type hypersensitivity) responses upon transfer to naïve animals [25]. Therefore, IDO expression in DCs in the nose-draining lymph nodes is essential for immune tolerance to inhaled antigens.

Tryptophan metabolism is altered in many allergic conditions and the IDO pathway plays a central role. Higher serum tryptophan concentrations are found in patients with seasonal AR [64] and asthmatic children [65]. Higher tryptophan and kynurenine levels are found in children with asthma and AR [26]. Low IDO activity has been found in asthma and AR patients [26,66]. IDO activity is induced by IFN-γ and is considered a Th1 cell activation marker [67]. During Th2 allergic inflammation, an elevated level of nitric oxide inhibits IDO activity by binding to the heme group of the enzyme. Therefore, the rationale of antioxidants as an anti-allergic therapy lies in their ability to block inducible nitric oxide synthase [67] and rescue the IDO activity which is essential to generate Tregs.

L-glutamine is another amino acid that plays a critical role in immune cell function. Although not an essential amino acid, L-glutamine is the primary fuel for immune cells and is essential for basic immune cell functions such as lymphocyte proliferation and cytokine production [68]. A recent study showed that abnormal glutamine metabolism is associated with allergic diseases [31]. IL-10-secreting B cells are a type of B regulatory cell that suppresses allergic reactions. Decreased numbers of regulatory B cells or functional changes in them are observed in patients with allergic disorders including AR, asthma, and AD [28,29,30]. The underlying mechanism of the defects in Bregs is the altered glutamine metabolism. In normal cells, glutamine is transported into the cells by a cell surface transporter called ASCT2 (alanine, serine, cysteine-preferring transporter 2), to be metabolized in a process called glutaminolysis [69]. B cells from patients with AR express low levels of ASCT2 and generate less IL-10^+^ regulatory B cells under IL-10-inducing culture conditions [31].

### 4.2. Dietary Lipids

The amount of dietary lipids and type of fatty acids influence allergic inflammation. High total fat, animal fat, saturated fatty acids (SFAs), cholesterol, *n*-6 polyunsaturated fatty acids (PUFAs), and medium-chain fatty acids (MCFs) are risk factors, whereas monounsaturated fatty acids (MUFAs) and *n*-3 PUFAs have protective properties. High animal fat and SFAs are associated with allergic rhinitis in human adults while high MUFA intake is associated with a lower risk for asthma [46,59]. In humans, high consumption of olive oil, a rich source of MUFAs, is associated with reduced risk for asthma in Italian adults [59] and teenagers in Taiwan [70].

A high-fat diet (60% Kcal from saturated fat) has been shown to increase serum TSLP in C57BL/6 mice and exacerbate dermatitis in mice through upregulation of TSLP in NC/Nga mice that develop AD spontaneously [18]. The high-fat diet increased TSLP in dorsal skin, infiltration of inflammatory cells, and epidermal thickening in NC/Nga mice compared with a low-fat diet. Dermatitis score was much lower in high-fat-fed NC-TSLP-KO mice, suggesting TSLP mediates a high-fat-diet-induced increase in dorsal skin inflammation [18]. Long-term feeding (10 months since weaning) of a Western diet (21.2% fat, 34% sucrose, and 0.2% cholesterol) also substantially increased spontaneously developed dermatitis in aged C57BL/6 mice, as compared with a control diet (5.2% fat, 12% sucrose, and 0.01% cholesterol) [71]. The Western diet-fed mice had increased epidermal thickness in their dorsal skin and much more epidermal hyperplasia in the lesion skin, with hypergranulosis and spongiosis typical of AD [71]. The Western diet leads to increased total bile acids, altered bile acid profiles, and elevated bile acid signaling through two bile acid receptors TGR5 (transmembrane G-protein-coupled receptor-5) and S1PR2 (sphingosine-1-phosphate receptor-2) in the lesion skin [71]. Lowering serum cholesterol with a bile acid sequestrant cholestyramine reduced epidermal hyperplasia and decreased Th2 and Th17 cytokines [71]. Therefore, dysregulated bile acid metabolites, induced by the Western diet, are the main contributors to the dermatitis lesion.

Besides saturated fatty acids and cholesterol, medium-chain fatty acids (MCFs) contained in coconut oil or palm oil also prove to be a dietary risk factor for allergy [5]. In a mouse model of peanut allergy, compared with *n*-6 PUFAs from peanut oil, MCFs decreased dietary peanut or OVA antigen absorption into the circulation and increased antigen in the Peyer’s patches, which resulted in a significant increase in activated DC cells [72]. Single feeding of peanut protein with MCFs resulted in increased serum IgE, anti-peanut IgG, and IL-13 production from splenocytes. MCFs promoted allergic sensitization through the upregulation of mRNA of TSLP, IL25, and IL-33 from jejunum epithelium and promoted Th2 cytokines in splenocytes in OVA-challenged mice. Moreover, MCFs also exacerbated orally challenged antigen-induced anaphylaxis compared with *n*-6 PUFAs.

The phospholipids isolated from asparagus (*Asparagus officinalis* L.) are demonstrated to have anti-allergic properties. Oral administration of these phospholipids suppressed serum total IgE and OVA-specific IgE in OVA-challenged mice and ameliorated clinical scores of AD induced by picryl chloride in NC/Nga mice [73]. Phospholipid and glycolipid fractions from asparagus also potently inhibited β-hexosaminidase release from cultured RBL-2H3 (rat basophilic leukemia-histamine-releasing cell line) cells, indicating a direct effect on degranulation in allergic responses [73].

Although conflicting results are generated from human studies about the effects of long-chain PUFA supplementation on asthma, AR, and AD [74], animal studies provide clear evidence of the protection of dietary *n*-3 PUFA in these allergic conditions. Dietary *n*-3 fatty acid α-linolenic acid shows beneficial effects in allergic inflammation by improving skin barrier function in AD mice [75] and attenuating symptoms in OVA-induced AR in mice, as compared with *n*-6 fatty acid linoleic acid [76]. Dietary linseed oil (enriched with α-linolenic acid) increases EPA-derived metabolite 15-HEPE (hydroxyeicosapentaenoic acid in eosinophils) in eosinophils in the nasal passage, which inhibits mast cell degranulation by binding to PPAR (peroxisome proliferator-activated receptor) γ [76]. In human mast cells, both EPA and DHA suppress IL-4 and IL-13 [77], suggesting their possible protective roles in type 2 inflammation. In contrast, long-chain *n*-6 fatty acid-derived arachidonic acid increases TNF-α and PGD_2_ in human mast cells [77], supporting the concept that an increased *n*-6/*n*-3 fatty acid ratio in the Western diet is pro-inflammatory and likely to promote type 2 inflammation. In the DNCB-induced AD mouse model, both dietary fish oil and fermented fish oil significantly alleviated scratching behavior, decreased epidermal thickness, and infiltration of cell infiltration in skin lesions, suppressed TSLP protein expression in ear tissue and serum histamine and IgE [19], with fermented fish oil having a better effect. Compared with natural fish oil, fermented fish oil resulted in higher TGF-β and IL-10 mRNA expression and a stronger suppressive effect on IL-13 and IFN-γ in the ear tissue due to higher content of EPA and DHA, known to be incorporated into the skin tissue. The suppressive effect on Th2 cytokines by fish oil and fermented fish oil may not be a direct effect on Th2 cells, but rather through the indirect effect of Tregs because fish oil does not affect Th2 differentiation [19]. Fermented fish oil did not increase Tregs in the spleens of these mice. However, increased Foxp3 expression in CD4^+^ T cells from fermented fish-oil-supplemented mice is observed upon anti-CD3/anti-CD28 activation, suggesting fermented fish oil alters the cytokine milieu to promote Treg differentiation. Additional research is needed to investigate the mechanisms of how EPA and DHA affect structural cells and innate immune cells to reduce type 2 allergic inflammation. Indeed, orally administered EPA was shown to markedly ameliorate special diet-induced AD-like symptoms in hairless mice accompanied by attenuated TSLP, IL-4, and IL-5, along with improved skin barrier function [78]. Analysis of the composition of lipids covalently bound to corneocytes revealed that dietary EPA significantly increased covalently bound ceramides in the stratum corneum [78].

Olive oil, as a major component of the Mediterranean diet, has many health benefits. Olive oil is enriched with monosaturated n-9 fatty acids. Recently, it was shown that olive oil confers protection again food allergies by improving gut mucosal barrier integrity [79]. Olive oil also enhances oral tolerance to dietary allergens by decreasing serum antigen-specific IgE, antigen-specific IgG, and histamine [80]. Increased IL-10 and decreased IL-4 associated with olive oil feeding indicate that Tregs and Bregs are induced. Detailed mechanisms warrant further investigation. Altered gut microbiota is also associated with an olive oil diet, and the polyphenols and other phytochemicals in olive oil may be the contributing factors. For example, uvaol, a triterpene in olive oil, exhibits anti-inflammatory activity in two murine models of allergic inflammation [81].

### 4.3. Dietary Fiber

Recent animal studies show that dietary fiber protects against AD or allergic asthma through its bacterial metabolites short-chain fatty acids, particularly butyrate [14,82,83]. Gut microbiota fermentation of dietary fiber into SCFAs is the key to the gut–skin axis or gut–lung regulation of allergic reactions in the skin and lungs. Consistent with animal studies, dysbiosis characterized by the enrichment of *Faecalibacterium prausnitzii* and a reduced capacity for butyrate fermentation in the human gut microbiome has been found in patients with AD [84]. Gut microbiota-derived butyrate has been found to be inversely associated with mite-specific IgE levels in childhood asthma [85]. Furthermore, infants who develop allergies in childhood have reduced bacterial enzymes for carbohydrate breakdown and butyrate production in their gut microbiome [86]. A recent clinical study in Japan showed that gut microbial factors are associated with AR [48]. The relative abundance of *Prevotella* was lower and the relative abundance of Escherichia was higher in AR patients compared with healthy controls [48]. *Prevotella* abundance reflects the intake level of dietary fiber and is linked to a diet based on plant foods. Decreased relative abundance of *Prevotella* is associated with the Western diet [87,88]. Increased abundance of *Escherichia* is linked to a high-protein diet [89]. A higher abundance of *Escherichia* is also found in children with asthma and rhinitis [90]. Despite observed alterations in the gut microbiota in allergic individuals, the efficacy of probiotic treatment remains unclear [91]. A more comprehensive approach, which restores the overall health of the gut microbiome through dietary approaches, might have better effects than the use of a single probiotic species. There is some evidence from human studies that a higher dietary fiber intake has protective effects on the clinical outcome of asthma [58,92,93].

Short-chain fatty acids, particularly butyrate, regulate type 2 inflammation mainly through the inhibition of HDAC (histone deacetylase) on various immune cells and structural cells. Vancomycin treatment in mice results in dramatic alterations in the gut microbiome characterized by decreased richness, diversity, and decreased abundance of butyrate-producing families, leading to increased susceptibility to allergic inflammation [83]. A supplement of SCFA in drinking water attenuated OVA or papain-induced allergic asthma by suppression of DC activation and trafficking, therefore restraining Th2 cell development in Peyer’s patches [83]. Butyrate also directly regulates ILC2 cells by suppressing IL-33-induced IL-13 and IL-5 production in cultured ILC2 lung cells from Rag2^−/−^ (recombination-activating gene 2 deficient) mice who lack T cells [24]. When administered either through drinking water or through an intranasal route, butyrate ameliorated ILC2 cell-driven lung inflammation. The inhibitory effect of butyrate on ILC2 cell proliferation was due to histone deacetylase (HDAC) inhibition [24]. In a mouse model of AR, intranasal administration of sodium butyrate improved clinical symptoms and nasal mucosal epithelial morphology, accompanied by decreased serum levels of Th2 cytokines and increased Th1 cytokines [94]. Butyrate attenuates TSLP protein expression level in stromal cells in nasal mucosa by working as an inhibitor of HDAC1 and HDAC3 [94]. Dietary fiber can influence asthma through epigenetic mechanisms by inhibiting HDAC enzymes [10]. Mouse studies showed that pups from pregnant mothers on a high-fiber diet or acetate are protected from house dust mite (HDM)-induced asthma [95]. Besides epigenetic regulation of HDAC, dietary fiber also affects the metabolism and function of structural cells at the barrier sites, which are critical for the initiation of an allergic reaction. For example, in a mouse model of HDM-induced AD, high-fiber (inulin, a highly fermentable dietary fiber) intake or butyrate protects animals from developing skin inflammation [14]. A lower disease severity is accompanied by an improved skin barrier, decreased epidermal thickening, less inflammatory cell infiltration, and decreased antigen-specific IgE. Butyrate feeding results in the enrichment of pathways related to immune and barrier function in skin transcriptome [14]. Surprisingly, butyrate does not modify skin immune cells before allergy exposure and does not affect skin Tregs. Butyrate blunts immune responses to HDM through enhancing mitochondria fatty acid β-oxidation and long chain fatty acid synthesis and promoting epidermal keratinocyte differentiation, therefore strengthening the skin barrier at the baseline and following HDM exposure [14].

Both the amount and type of dietary fiber affect susceptibility to allergic airway inflammation and the severity of the inflammation. A low-fiber diet (<0.3%) increases susceptibility to HDM-induced allergic airway inflammation in mice compared with the standard 4% chow diet [14]. Besides increased eosinophils and lymphocytes in the lung, elevated total IgE and HDM-specific IgG_1_ were observed in mice on a low-fiber diet compared with a normal-level fiber diet, suggesting that the low-fiber diet promotes systematic allergic inflammatory responses. The low-fiber diet also results in a more activated phenotype of dendritic cells, as indicated by increased surface expression of CD40, CD80, PD-L1, and PD-L2. A high-pectin (a water-soluble and highly fermentable dietary fiber) diet decreases susceptibility to allergic airway inflammation, as compared with a high-cellulose (a water-insoluble dietary fiber which is not fermented by the gut microbes) diet, indicating the gut fermentation process of pectin to SCFA, particularly propionate, is the key for this beneficial effect [14]. Nonetheless, even the high-pectin diet does not increase SCFA levels in the lung. High pectin intake leads to increased propionate in the circulation which enhances bone marrow hematopoiesis and generation of DC precursors, which express low levels of MHCII and CD40 and have an impaired ability to promote Th2 cell responses.

Highly fermentable fibers other than pectin also influence allergic inflammation, an effect dependent on the gut microbiota fermentation process. Compared with a high-fiber diet composed of cellulose, a high-inulin or high-psyllium diet induces increased serum bile acids and triggers eosinophilia in the colon and lungs [13]. Increased bile acids bind to farnesoid X receptors on stromal cells and epithelial cells and trigger the release of IL-33, which acts on ILC2 cells to produce IL-5, therefore promoting allergen-induced type 2 barrier inflammations in the lungs [13]. This effect of inulin is dependent on intestinal bacterial bile salt hydrolase (BSH) expressed on *Bacteroides ovatus* which hydrolases conjugated bile acids into unconjugated bile acids. Inulin promoted the growth of *Bacteroides ovatus*, therefore leading to increased serum bile acids.

### 4.4. Dietary Flavonoids and Other Phytochemicals

Flavonoids are a major type of phytochemicals in the diet and are naturally occurring phenolic compounds which are commonly found in fruits, vegetables, herbs and spices, legumes, tea, and vinegar [96,97]. There are six subclasses of dietary flavonoids based on their chemical structures, namely flavanols, flavones, isoflavones, flavanones, flavonols, and anthocyanidin [96,97]. Accumulating evidence has shown the anti-allergic effect of dietary flavonoids. The effects of dietary flavonoids in AR, AD, and asthma are summarized in Table 1.

As a major dietary flavonol-type flavonoid, quercetin is found in many fruits and vegetables including onions, shallots, apples, berries, tea, tomatoes, grapes, nuts, and seeds. The anti-inflammatory effect of quercetin is well documented in various animal models of allergy [98]. Quercetin is effective in reducing allergic symptoms by decreasing serum IgE and Th2-related cytokines, reducing eosinophil, neutrophil, and mast cell infiltration into local tissue, reducing epithelial thickness in the lung and hyperkeratosis, and suppressing epithelial cell-derived cytokines IL-25, IL-33, and TSLP [98]. However, in most in vivo animal studies, quercetin is administered through i.p. injection. As quercetin is a glycone (namely, carbohydrate conjugate), how dietary quercetin is metabolized by the gut microbiota and the subsequent effects on allergic inflammation remain to be explored. In a recent study, oral administration of quercetin was shown to attenuate nasal symptoms of OVA-induced AR in BALB/c (Halsey J Bagg albino mice strain c) mice by suppressing angiogenic factors and proinflammatory cytokines TNF-α, IL-6, and IL-8 in nasal lavage fluids [99]. The minimum effective dose for the above in vivo inhibition is similar to the maximum daily recommended dosage for dietary quercetin supplements. Furthermore, in IgE-sensitized mouse peritoneal mast cells, quercetin at concentrations comparable to physiological blood concentrations achieved by recommended dietary quercetin supplement intake dosage completely inhibited VEGF (vascular epithelial growth factor) and bEGF (basic fibroblast growth factor) at mRNA level and potently suppressed TNF-α, IL-6, and IL-8 at mRNA level [99]. In human keratinocytes treated with a cytokine cocktail that induces TSLP production, quercetin suppressed TSLP production and MMP mRNA expression [20]. Quercetin also increased protein expressions of epithelial junction protein E-cadherin, Occludin, and two proteins related to tissue repair: Twist and Snail [20], indicating quercetin’s ability to promote wound repair. Notably, quercetin also highly upregulated IL-10 mRNA and further increased IL-10 following proinflammatory cytokine cocktail treatment, indicating that quercetin affects the cytokine milieu in the tissue to promote IL-10 T or B regulatory cells under inflammatory conditions. Baicalin, a flavone-type flavonoid present in lettuce and cantaloupe, also regulates IL-10/IL-17 and is able to attenuate symptoms in a mouse model of AR [100].

**Table 1 nutrients-15-03683-t001:** Beneficial effects of dietary phytochemicals in allergic diseases.

Flavonoids	Experimental Models	Results	Reference
Quercetin	OVA-induced AR in BALB/c mice 25 mg/kg dosage 5 d during challenge	Inhibited sneeze and nasal rubs	[99]
		Suppressed angiogenic factors	
		and TNF-α, IL-6, IL-8	

Quercetin	Human HaCaT keratinocytes	Promoted wound repair	[20]
		↑ E-cadherin, Occludin, Twist, Snail	
		↑ IL-10 at basal level	
		↓ MMP1, MMP2, MMP9, ↓ TSLP	
Kaempferol	DNCB/mite extract induced	↓ ear thickness	[101]
	dermatitis in BALB/c mice ear	↓ Dermal and epidermal thickness	
	15, 50 mg/kg 5 d on/2 d off	↓ Mast cell infiltration	
	for 4 wks following 2nd DNCB	↓ Serum IgE	
		↓ mRNA of IL-4, IL-13, IFNγ	
		IL-17a, IL-6, IL-31, TSLP	
		in ear tissue	
	Jurkat cells	↓ αCD3/CD28, PMA/A23187	
		stimulated IL-2 production	
		↓ AICD	
		Inhibited MRP-1 activity	
		Suppressed JNK phosphorylation	
Kaempferol	OVA-induced allergic asthma	↓ TGF-β production in the lung	[53]
	in BALB/c mice	↑ E-cadherin and epithelial thickening	
	10, 20 mg/kg for 3 days	↓ α-SMA,	
	during challenge	↓ Collagen IV, ↓ MT1-MMP	
		↓ Lung fibrosis	
		↓ PAR1 signaling	
Naringenin	OVA-induced AR in Sprague Dawley rats	Reduced nasal scratching and number of sneezing	[102]
	100 mg/kg 7 d during challenge	Decreased serum IL-4, IL-5	
Diosmetin	DNCB-induced AD	↑ Skin barrier function	[103]
	in SKH-1 hairless mice	↓ Skin swelling, erythema	
	5 mg/kg for 14 d	↓ Skin erosion and dryness	
	during challenging period	↓ Epidermal thickness	
		↓ Mast cell infiltration in skin	
		↓ Serum IgE and IL-4	
Baicalin	OVA-induced AR	Reduced inflammatory cells	[100]
	in BALB/c mice	in nasal lavage fluid	
	L-Baicalin 50 mg/kg	↓ Nasal symptoms	
	H-Baicalin 200 mg/kg	↓ Thickness of nasal epithelium	
	10 d following sensitization	↓ Nasal mucus production	
	and 4 d before challenge	↓ IL-17, ↑ IL-10 in nasal discharge	
		↓ OVA-specific IgE, IgG1 antibodies	
		Inhibited autophagy in nasal mucosa	
Baicalin	DNTB-induced AD	↓ Dorsal skin thickness	[104]
	in BALB/c mice	↓ Trans-dermal water loss	
	50, 100, 200 mg/kg	↓ Epidermal thickness	
	14-d following DNTB stimulation	↑ Skin barrier function, ↓ TSLP	
		↓ NF-κB signaling pathway in skin	
		↓ JAK, STAT signaling pathway	
		↑ Actinobacteria	
Licoricidin	DNCB/mite induced atopic	↓ Epidermal and dermal tissue	[105]
	dermatitis in ear tissue	↓ Infiltrating mast cells	
	in BALB/c mice	↓ Serum IgE, IgG1, IgG2a	
	50 mg/kg 5 d on/2 d off following	↓ mRNA of IL-4, IL-5,	
	the 2nd DNCB for 4 wks	IL-6, IL-13 in ear tissue	
		↓ Size and weight of draining	
		lymph nodes	
		↓ T cells and Th2 cytokines in dLNs	
		↑ T cell PTPN1 phosphorylation in dLNs	
		↓ DC activation through	
		antagonizing PTPN1	
Resveratrol	3-month repeated OVA	↓ Airway hyperresponsiveness	[52]
	exposure induced chronic	↓ Inflammatory cells, IL-4, Il-5, Il-13	
	asthma in BALB/c mice	in BAL fluid	
		↓ Lung infiltration of inflammatory cells	
		↓ Goblet cell number	
		↓ Peribronchial α-SMA	
		↓ Collagen amount in lung tissue	
SDG	OVA-induced AR	Ameliorated sneezing number	[106]
	in BALB/c mice	Decreased eosinophil and neutrophil	
	100 mg/kg 3 times a week for	infiltration	
	4 wks before initial sensitization	Enhanced β-glucuronidase	
		activity and increased	
		ED levels in nasal passage	

HACAT—cells-human epidermal keratinocyte cell line; MMP—matrix metalloproteinases; DNCB—dinitrochlorbenzene; Jurkat cells–T-lymphocyte cell line; CD—cluster of differentiation; PMA—phorbol-myristate-acetate; AICD—activation-induced cell death; MRP—motility related protein; JNK—c-Jun-N-terminal kinases; TGF—transforming growth factor; MT1-MMP—membrane type 1-matrix-metalloproteinase; OVA—ovalbumin; SDG—secoisolariciresinol diglucoside; ED—enterodiol; PTPN1—protein tyrosine phosphatase-receptor type 1; dLN—draining lymph nodes; alpha SMA—anti-alpha-smooth muscle actin; PAR—protease-activated receptor. ↑, up-regulation; ↓, down-regualtion.

Kaempferol, another flavonol-type flavonoid found in many fruits, vegetables, herbs, teas, and medicinal plants, also exhibits anti-inflammatory, antioxidant, and anti-allergic properties. In cultured lung epithelial BEAS-2B (human broncho-epithelial-alveolar stem cell-derived cells) cells, nontoxic kaempferol suppresses LPS (lipopolysaccharide)-induced TGF-β production, TGF-β-induced myofibroblast formation, LPS-induced collagen, and MT1-MMP, suggesting its ability to suppress the epithelial-to-mesenchymal transition and fibrosis. In a mouse model of asthma, orally administered kaempferol not only suppressed eosinophil infiltration and airway inflammation but also inhibited the airway epithelial-to-mesenchymal transition (EMT) and fibrosis [53]. As fibrotic airway remodeling is characteristic of asthma, leading to lung function deterioration, and is not treated by current drug therapy, kaempferol may be a potential therapy for asthma-related airway construction and is worthy of further clinical studies. Kaempferol also protects mice against AD by suppressing T cell activation though interaction with MRP-1 [101].

Oral administration of naringenin, a flavanone mostly found in citrus peel, was shown to significantly reduce nasal scratching score in rats with OVA-induced AR with improved histology in the nasal epithelium and decreased serum IgE, IL-4, and IL-5 [103]. In addition, naringenin inhibited TSLP production in PMA/Ionophore-activated human mast cells (HMC-1 cells) through inhibition of NF-κB and TSLP-induced mRNA expressions of IL-13, TNF-α, IL-17 receptors, and TSLP receptors in these cells [102]. Therefore, naringenin and many other flavonoids may have a protective role against allergic conditions in allergen-sensitized individuals by regulating TSLP, the key initiator of Th2-driven allergic inflammation. Future clinical studies of naringenin on human allergic conditions are warranted.

The gut microbiota-derived metabolites are critical for the anti-allergic function of some flavonoids. For example, the flavone glycoside diosmin and its aglycone form diometin were shown to diminish DNCB-induced AD symptoms in SKH-1 hairless mice, such as increased trans-epidermal water loss and hydration, epidermal thickness, and infiltration of mast cells [103]. Decreased serum IgE and IL-4 in these mice were observed for both diosmin and diometin; however, in cultured RBL-2H3 cells, only diosmetin and not diosmin showed inhibitory effects on IL-4 production. This suggests that the in vivo anti-allergic effect of diosmin depends on its breakdown into the aglycone form by the gut microbiota. The anti-AD effect of baicalin also depends on the gut microbiota because fecal transplantation from baicalin-treated mice to GF (germ-free) mice resulted in significantly reduced skin thickness and clinical symptoms accompanied by decreased serum IgE and IL-4 [104].

Some dietary phytochemicals other than flavonoids also exhibit strong anti-allergic properties. Licoricidin, a component isolated from licorice (*Glycyrrhiza uralensis*) root which is a commonly used herb in traditional medicine, shows protection against mouse AD by suppression of T cell activation through regulating PTPN1 activity [105]. Resveratrol, the best-studied polyphenol, inhibits mast cell activation and shows potential in treating allergic conditions [107]. A recent study showed that orally administered resveratrol inhibits airway inflammation and remodeling in a murine chronic asthma model [52]. In mice that developed asthma from repeated exposure to OVA over the course of three months, resveratrol effectively inhibited TGF-β production and signaling in the lung tissue and epithelial–mesenchymal transition, therefore improving lung function as measured by airway hyper-responsiveness to methacholine [52]. This suggests the potential of resveratrol as an effective therapy for treating airway remodeling associated with asthma. The nasal metabolism of phytoestrogen is important in the observed anti-allergic property for secoisolariciresinol diglucoside (SDG), a phytoestrogen enriched in flaxseed. Dietary SDG was shown to ameliorate OVA-induced AR symptoms in mice and was associated with less infiltration of neutrophils and eosinophils [106]. SDG did not alter antigen-specific IgE or IgG levels in plasma. Enterodiol (ED), the bacterial metabolite of SDG, is circulated in the blood in the form of EDGlu, but converted to ED aglycone in the nasal passage where it inhibits IgE-mediated degranulation of basophil degranulation in a GRR (interferon-gamma response region) 30-dependent manner [106]. Host or nasal cavity microbiota-derived β-glucuronidase activity is responsible for generating active phenolic metabolites. The metabolites form many phenolic compounds, including resveratrol, EGCG (epigallocatechin gallate), and curcumin, and are likely able to control the nasal local tissue environment in a similar manner to reduce effector immune cell activation in allergic responses. Several plant extracts show good anti-allergic abilities in several animal studies, although the exact active chemicals in these extracts remain to be determined. For example, the anti-AD effect of celery extract [108], black soybean extract [109], and the anti-AR effects of the extracts of *Musa paradisiaca* L. inflorescence [110], *Piper nigrum* fruit [111], and *Cuminum cyminum* L. seed [112], are recently demonstrated in various animal models. Exploring dietary phytochemicals and their metabolites for anti-allergic potential represents a new direction for basic research and more clinical studies are needed to verify their effects in human patients. The beneficial effects of dietary phytochemicals [51,113] in allergic diseases are supported by the recent clinical intervention studies listed in Table 2. Daily intake of 15 g of a novel barley-based formulation for 14 days proved to significantly reduce all symptoms in patients with AR and, with even better results than fexofenadine in terms of controlling nasal congestion, postnasal drip, and headache [114]. This beneficial effect on the control of allergic symptoms could be due to the phytochemicals and soluble fiber present in the barley drink power.

### 4.5. Vitamins and Minerals

Vitamins and minerals have long been known for their immunomodulatory roles. Vitamins A, D, and E, and trace elements zinc and iron, are particularly important dietary factors, influencing allergic inflammation and the development of allergic diseases. Sufficient intake of Vitamins A, D, and E is required to control asthma [5]. Supplementation with vitamins E and D alone or in combination improves symptom management of AD [116]. Serum vitamin D level is a determining factor in remission with standard therapy for AD. A serum level of 1, 25(OH)_2_VD_3_ higher than 20 ng/mL plus standard therapy is sufficient to reduce the severity of AD [115]. In a randomized, double-blind, placebo-controlled clinical study, an oral supplement of 5000 IU/day vitamin D_3_ in patients with AD significantly increases the serum level of 1, 25(OH)_2_VD_3_ to a much higher level than the placebo group, and this dosage achieved sufficiency in 100% of the patients [115]. Vitamin D also shows potential in managing airway remodeling in asthma, based on a number of in vitro studies showing the inhibitory effects of vitamin D on bronchial smooth muscle cells, human airway smooth muscle cells, human asthmatic bronchial fibroblasts, and human bronchial fibroblasts [54]. Recent studies suggest that deficiencies in iron, zinc, and vitamins contribute to the etiology of atopic diseases in children, and supplementation with micronutrients is considered essential for managing the atopic march [50]. Even in adults, evidence also accumulates to support the role of micronutrients in the etiology or treatment of atopic diseases [117]. At cellular and molecular levels, micronutrients are essential for the proper growth and function of all immune cells. Vitamins A and D are particularly important in maintaining immune tolerance to allergens by promoting Treg induction [5]. Deficiencies in micronutrients mimic pathogen infection and lead to the activation of immune cells, therefore priming the host for a Th2 response when encountering an allergen [50]. For example, iron depletion is related to elevated IgE levels and functional iron deficiency is sufficient to evoke mast cell degranulation [50]. Therefore, adequate intake of micronutrients contributes to immune tolerance by increasing allergic resilience, promoting Tregs, and maintaining Th1/Th2 balance.

Recent evidence suggests that vitamin E plays a role in AR. In a mouse model of OVA-induced AR, oral administration of vitamin E (100 mg/kg/day) at the time of OVA sensitization decreased bronchoalveolar lavage fluid (BALf) IL-33 (more than 50%), IL-25, and Th2 cytokines IL-4, IL-5, and IL-13 [118]. Interestingly, co-administration of selenium resulted in a further decrease in IL-13 production, indicating synergistic effects between vitamin E and selenium on IL-13 production. Vitamin E also decreased serum IgE by more than 50% and histamine by 78% [118]. In a similar mouse model of AR, nasally administered α-Tocopherol before nasal challenge in OVA-sensitized mice suppressed nasal symptoms, with fewer inflammatory lesions and better integrity in nasal tissue [119]. Reduced nasal eosinophils and mast cells, upregulated Th1 cytokine IFN-γ gene expression and downregulated Th2 cytokines IL-4, IL-5, and IL-13 gene expression, and reduced total IgE, specific IgE, IgG, and the PI3K-PKB (phosphatidylinositol 3-kinase-protein kinase) pathway in mast cells were observed in α-Tocopherol-treated mice. These results suggest that vitamin E status (systemic or local) affects both arms of the innate and adaptive immune responses in AR. The molecular mechanisms of how vitamin E affects epithelial cells and ILC cells warrant further research.

The trace element zinc is essential for immune function. Zinc deficiency is often linked to allergies. A zinc supplement is shown to be effective in relieving asthma but not beneficial to AD [50,57]. In an animal asthma model, zinc deficiency is related to greater airway hyper-responsiveness compared with normal zinc intake, whereas zinc supplementation reduces inflammatory cell infiltration and improves clinical symptoms [120]. At the cellular level, the beneficial impact of zinc on allergic immune reactions mainly includes T cell differentiation and antigen-specific T cell proliferation. In cultured human PBMCs (peripheral blood mononuclear cells), zinc deficiency increases Th17 differentiation [121]. On the other hand, the zinc supplement in the cell culture of allergen-stimulated PBMCs alters the Th1/Th2 ratio and decreases the proportion of Th17 [122]. Zinc supplementation also enhances Treg differentiation either in allergen-stimulated PBMCs from atopic patients [37] or in TGF-β treated PBMCs and mixed lymphocyte cultures [36]. Moreover, in vitro, a supplement of zinc suppresses allergen-stimulated proliferation of atopic PBMCs [37].

Iron is another trace element that has been linked to the etiology of atopic diseases [123]. As the most common nutritional disorder, iron deficiency is associated with half anemia which affects about a third of the world’s population [124]. Iron deficiency can be present either as low hemoglobin levels in the blood or with low levels of metabolically active iron despite normal ferritin iron storage in the body [123]. While the majority of the iron requirement in the human body is met by recycling from senescent red blood cells by splenic macrophages and redistribution to other cells, dietary intake of iron provides only about one-tenth of the daily requirement [123]. Therefore, the macrophage regulation of the iron pool and metabolism is highly important, which determines the activation state of the immune system. When iron mobilization is blocked under various conditions, iron deficiency in immune cells is perceived as a danger signal and leads to abnormal activation such as mast cell degranulation [123]. Hepcidin, an acute-phase protein induced by inflammation, affects the iron level in the circulation by blocking iron absorption and iron mobilization from macrophages, thereby leading to functional iron deficiency in atopic patients [123]. Raw milk whey-protein-derived β-lactoglobumin, as a carrier of iron flavonoid complexes, has been shown to be effective in delivering iron to human monocytic cells and impairing antigen presentation of allergens [43]. The so-called holo β-lactoglobulin complexed with ligands is able to reduce allergic symptoms in mice by decreasing lymphocytic and B cell proliferation and promoting Treg induction [42]. Consistent with preclinical observations, in a randomized, double-blind, placebo-controlled study (*n* = 51), a 6-month course of supplementation with a β-lactoglobulin-based micronutrients lozenge formula (iron, polyphenol, retinoic acid, and zinc) in grass/birch pollen allergic women resulted in more improvement in nasal symptoms, as compared with the placebo group (42% vs. 13%) in an allergen-independent manner [49]. Dietary intervention with the lozenge significantly improved iron status in myeloid cells, as indicated by increased hematocrit levels and reduced width of red cell distribution, and increased iron levels in CD14^+^ monocytes, but not in T lymphocytes. This study highlights the importance of iron deficiency in allergy development, and correcting micronutrient deficiency in immune cells as an effective therapy for allergy treatment.

Copper is closely related to iron metabolism. The copper-containing ferroxidase ceruloplasmin is involved with iron mobilization during acute inflammation, and its elevation indicates iron deficiency [117,123]. A recent clinical study in Japan showed that multiple nutritional and gut microbial factors are associated with AR [48]. Four nutrients (retinol, vitamin A, cryptoxanthin, and copper) were negatively associated with AR [48]. In a cohort study in Poland (*n* = 80), the plasma level of Cu was found to be associated with AR in children aged 9–12 [125].

Selenium is an essential trace element that is very important for optimal immune function. Populations from China, the UK, and Scandinavia generally tend to have reduced Se levels [126]. While Se deficiency leads to impaired immune responses, Se supplements boost immune competence. Selenium is an essential component of glutathione peroxidase (GSH-Px), a key antioxidant enzyme that functions to reduce peroxides, therefore protecting against inflammation-induced, excessive oxidative stress-related membrane damage [127]. While a lower serum level of selenium is reported to be associated with an increased risk of asthma in human studies [128,129], an animal study demonstrated that a lower level of selenium is associated with a lower asthma outcome. Although adequate dietary intake of selenium does not protect against the development of allergic asthma in mice, dietary selenium supplements have a synergistic anti-asthma effect with vitamin E in reducing airway inflammation and Th2-related cytokines [118]. In a mouse model of OVA-induced AR, co-administration of selenium with vitamin E resulted in a further decrease in IL-13 levels, as compared with supplementation with selenium or vitamin E alone [118], indicating that selenium and vitamin E affect different pathways of IL-13 production.

## 5. Obesity and Allergy

Due to the increasing prevalence of obesity and allergic diseases worldwide in recent decades, the link between obesity and individual allergic disease is of great interest. Obesity is a proven risk factor for asthma [130,131,132] and negatively impacts asthma outcomes [133]. Previously, no clear association was made between obesity and allergic rhinitis [130,131]; however, a recent meta-analysis study showed that obesity is perhaps associated with a higher risk of allergic rhinitis in children [134]. Moreover, obesity can contribute to the exacerbation of inflammation in severe persistent allergic rhinitis through increased IL-1β and leptin levels [135]. A growing body of evidence suggests a link between obesity and atopic dermatitis [136]. Although the prevalence of atopic dermatitis is higher in obese children and adults, the association between obesity and the severity of atopic dermatitis varies with age and gender [136]. The proposed underlying mechanisms for the link between obesity and allergy include pro-inflammatory adipokines (leptin, IL-6, TNF-α) released from adipose tissue [133], pro-inflammatory Th1 cells and Th17 cells associated with adipose tissue from obese individuals [5], and the ILC2–eosinophil–macrophage axis [5] in adipose tissue.

Dietary interventions producing weight loss in obese patients have been shown to be effective in improving asthma control [137]. Randomized controlled trials on dietary intervention showed that weight loss through restrictive diets with low energy is effective in improving asthma outcomes [138] and reducing airway inflammation in obese patients [139]. Even a normal caloric diet with a reduced content of fat, particularly saturated fat, was associated with reduced body weight and improvement of asthma-related quality of life in obese pubertal adolescents [140]. Although there are very limited studies, weight loss is associated with improved symptoms in atopic dermatitis. In a case report, weight loss through combined dietary control and exercise treatment improved skin lesions and normalized IgE and eosinophil counts in an obese patient who did not respond to standard cyclosporine treatment [141]. A randomized controlled study showed that weight reduction in obese patients with atopic dermatitis was associated with significant improvements in symptoms of atopic dermatitis, measured by eczema area and severity index score and decreased dosage of cyclosporine [142]. There has been no study on the effect of dietary intervention-induced weight loss on allergic rhinitis.

Plant-based diets are effective for weight loss [143,144,145] and can be an effective strategy for weight control, as well as in the treatment of obesity [145]. A plant-based vegan diet excludes all animal products, mainly consisting of grains, legumes, and vegetables and fruits; while in comparison, a vegetarian diet does not eliminate all animal products but emphasizes the consumption of fruits, vegetables, and nuts [145]. The weight reduction effect of such diets may be attributed to reduced calories and low fat intake [145]. Plant protein, as part of a plant-based diet, has recently been shown to be a contributing factor for weight control in overweight individuals [143]. An increased intake of protein and a decreased intake of animal protein are associated with a decrease in body fat mass. Plant-based diets are nutritionally adequate if planned well [144]. However, nutrient intake in the long term can be a concern, as revealed in a study of the weight-loss effects of a vegan diet in overweight postmenopausal women. The adoption of a low-fat vegan diet for 14 weeks leads to changes in macronutrients such as decreased intake of total fat, saturated fat and cholesterol, protein, and increased carbohydrate and fiber intake [144]. In terms of micronutrients, the vegan diet increased intakes of total vitamin A, β-carotene, thiamine, vitamin B6, folic acid, vitamin C, magnesium, and potassium, but decreased intakes of vitamin D, vitamin B12, calcium, phosphorous, selenium, and zinc [144]. Fortified food or supplements may help those following a vegan diet to meet the requirements of micronutrient intakes.

Despite limited data being available, plant-based diets appear to be remarkably effective in improving asthma [146] and atopic dermatitis [147]. According to a report from Sweden, a vegan diet therapy has a pronounced favorable effect on bronchial asthma [146]. After following the diet therapy for one year, patients became more tolerant of various environmental stimuli, such as dust, smoke, and flowers [146]. A significant decrease in asthma symptoms and improvement in clinical variables resulted in reduced needs for medication [146]. Similar striking results show that a two-month course of treatment with a customized vegetarian diet strongly inhibited the severity of atopic dermatitis [147]. A sharp reduction in the number of peripheral eosinophils and of PGE_2_ (prostaglandin E_2_) synthesis by monocytes was associated with this treatment [147]. Body weight-independent mechanisms with these diets may contribute to the observed beneficial effects on allergy outcomes, in addition to efficacy in body weight loss. In contrast to the Western diet which contains high amounts of pro-inflammatory nutrients, plant-based diets are enriched with micronutrients and dietary flavonoids associated with potent anti-inflammatory and anti-allergy effects (Figure 2). A plant-based diet may be particularly useful for the treatment of severe allergic diseases associated with obesity. Further clinical studies are required to validate the speculation.

## 6. Conclusions

In conclusion, diet and nutrition play a key role in the development and severity of allergic diseases by regulating tissue and immune homeostasis. Excessive calories, high intake of protein and saturated fatty acids, or lack of dietary fiber and micronutrients can trigger the defense mechanism in the immune system and prime the host for allergic reactions. Therefore, calorie restriction, coupled with sufficient dietary fiber and adequate macronutrient intake, will be essential for maintaining immune tolerance to allergens. The plant-based diets, which emphasize the high consumption of fruits and vegetables, grains, and legumes while avoiding or reducing animal foods, are associated with the reduction of inflammation and weight loss. Further dietary intervention studies are warranted to explore the potential beneficial effects of plant-based diets and the specific nutrients related to such diets on allergic outcomes. As basic research efforts identify more novel dietary components with anti-allergic properties, randomized placebo-controlled trials are also needed to verify their efficacy in human patients. Nutritional therapy holds great promise in reducing allergy symptoms, either as primary therapy and treatment or in support of drug therapy. Assessment of nutritional status and anthropometric characteristics of the patients, and analysis of host and gut microbiota by the multi-omics approach, will be important in future clinical trials to identify novel mechanisms linking nutrition and allergy.

## Figures and Tables

**Figure 1 nutrients-15-03683-f001:**
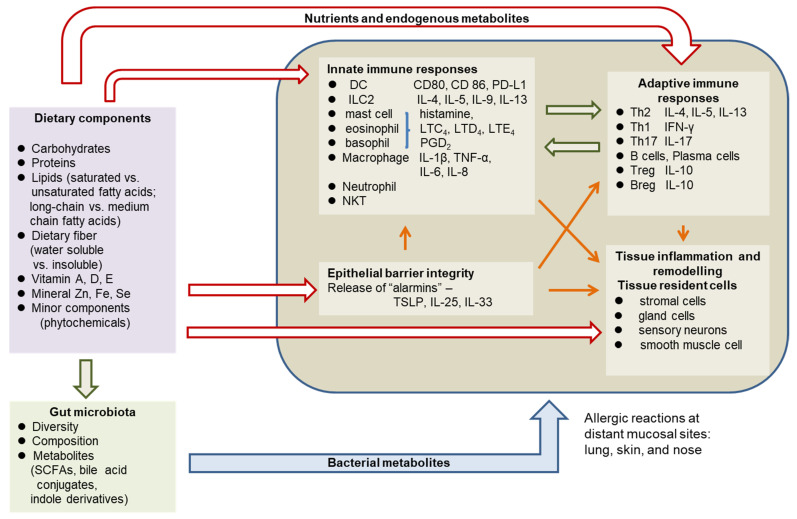
The impact of diet and nutrition on allergic reactions in the lungs, skin, and nose. The arrows indicate regulation. Red arrows represent nutrients and endogenous metabolites and blue arrow represents bacterial metabolites. Food components and endogenous metabolites can affect all stages of an allergic reaction by influencing the epithelial barrier and the release of alarmins, by interacting with innate and adaptive immune cells though special receptors to either promote immune activation or induce tolerance, and by directly acting on tissue epithelium and resident cells to regulate tissue inflammation and remodeling. Diet plays a critical role in determining the ecology of the gut microbiota including diversity, composition, and metabolism. Bacterial metabolites can also reach distant organs and regulate all these processes through multiple mechanisms. DC: dendritic cells; ILC2, type 2 innate lymphoid cells; TSLP, thymic stromal lymphopoietin; SCFAs: short-chain fatty acids; LTC_4_, leukotriene C_4_; LTD_4_, leukotriene D_4_; LTE_4_, leukotriene E_4_.; PGD_2_, prostaglandin D_2_; NKT: natural killer T cells; Treg, T regulatory cells; Breg, B regulatory cells.

**Figure 2 nutrients-15-03683-f002:**
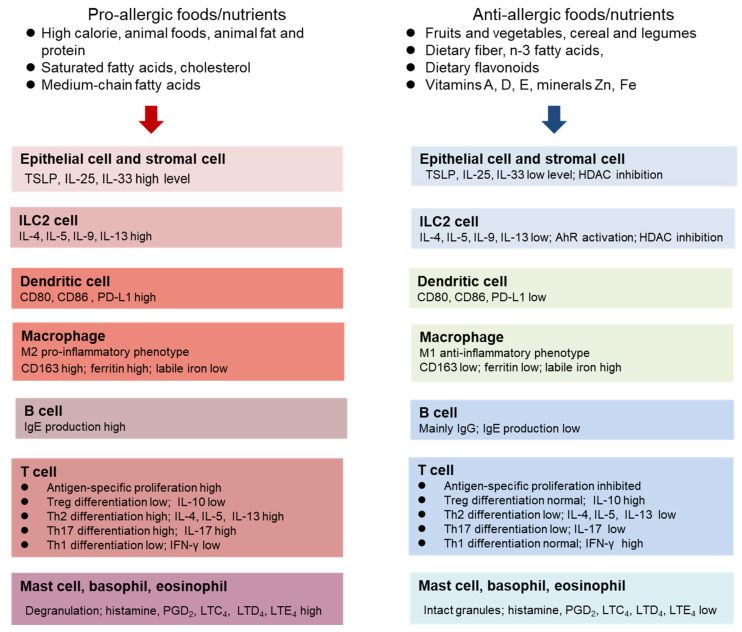
The roles of nutrients and foods in allergic inflammation. Epidemiological, clinical, and animal studies have demonstrated that the Western diet promotes allergy and exacerbates symptoms of allergic diseases, whereas nutritionally balanced plant-based diets protect from allergy and reduce the severity of allergic diseases. The pro-allergic nutrients associated with a Western diet promote the production and release of TSLP, IL-25, and IL-33 from epithelial cells and stromal cells and activate ILC2 cells to produce large amounts of IL-4, IL-5, IL-9, and IL-13, therefore producing a cytokine milieu for type 2 allergic inflammation reactions characterized by aberrant IgE and type 2 cytokines. By contrast, plant-based diets contain high amounts of anti-allergic nutrients which can suppress type 2 allergic inflammation through inhibition of type 2 cytokine production in ILC2 cells via activation of AhR, promotion of the generation of tolerogenic dendritic cells, anti-inflammatory macrophages, and Tregs, and suppression of the release of histamine, prostaglandins, and leukotrienes from granulocytes. AhR, aryl hydrocarbon receptor; ILC2, innate lymphoid cells; Treg, T regulatory cell; TSLP, thymic stromal lymphopoietin; PGD_2_, prostaglandin D_2_; LTC_4_, leukotriene C_4_; LTD_4_, leukotriene D_4_; LTE_4_, leukotriene E_4_. HDAC, histone deacetylase.

**Table 2 nutrients-15-03683-t002:** The impact of dietary supplements in allergic diseases.

Year Location	Study Design	Subjects and Intervention	Results	
2022	RCT	Patients (*n* = 60) with	Improved allergic symptoms	[113]
Tokyo		eye/nose allergic symptoms	including eye itching,	
Japan		Supplementation of	sneezing, nasal discharge,	
		200 mg quercetin for 4 wks	sleep disorder	
		vs. the placebo food	↓ Nasal discharge ecosipophil	
			Improved life quality	
2022	RCT	AR patients (*n* = 16)	↑ Overall symptoms in	[51]
Chiang Mai		Treatment with10 mg cetirizine	62.5% in shallot group	
Thailand		for 4 wks plus oral supplement	37.5% in placebo group	
		of 3 g shallot capsule vs.	↓ Overall symptom score	
		the placebo capsule	↓ Total ocular symptom score	
2022	RCT	AR patients (*n* = 77)	Improved all symptoms	[114]
Tehran,		Treatment with 60 mg	except cough in both groups	
Mashhad		Fexofenadine (FX) for 14 d.	MS better in nasal congestion,	
Iran		vs. 15 g dried power of,	postnasal drip, and headache	
		Ma-al-Shaeer (MS),	↓ Serum total IgE in both groups	
			a barley-based hot-water	
			extracted formulation	
2022	RCT	Allergic women (*n* = 51)	↓ Total nasal symptom score	[49]
Vienna,		Supplement for 6-month of	42% improvement in treated	
Austria		a lozenge called holoBLG (*n* = 25)	group vs. 13% in placebo group	
		containing β-lactoglobulin with	45%, 31%, 40% improvement in	
		iron, polyphenol, retinoic acid,	combined symptom score in	
		zinc vs. placebo (*n* = 26)	holoBLG group in birch peak,	
			entire birch season, the entire	
			grass pollen season	
			↑ Iron levels in circulating	
			CD14^+^ monocytes	
			↑ Hematocrit values	
			↓ Red cell distribution width	
2018	RCT	Patients with AD (*n* = 65)	↑ Serum vitamin D level	[115]
Mexico City		Standard treatment with	Inverse relationship between	
Mexico		Vitamin D3 5000 IU/day	final serum vitamin D level	
		for 12 wks vs. no extra vitamin	and severity of AD	
			Serum vitamin D > 20 ng·/mL	
			with standard therapy is sufficient to	
			reduce AD severity	
2019	RCT	Asthma patients (*n* = 17)	Inulin decreased airway	[92]
Newcastle		Treated with 7 d inulin	eosinophils and HDAC9	
Australia		(6 g powder twice daily),	expression in sputum cells	
		inulin + probiotic, placebo	Inulin improved asthma	
		with a 2 wks run-in and	control in poorly controlled	
		2 wks wash out periods	eosinophilic asthmatics	

RCT, randomized controlled trial; HDAC, histone deacetylase; ↑, up-regulation; ↓, down-regualtion.

## Data Availability

Data sharing is not applicable to this article. No new data were created in this study.

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
