# Peer review of "The Role of Diet and Nutrition in Allergic Diseases"

_nutrients, 2023, doi:10.3390/nu15173683_

Round 1
Reviewer 1 Report
Dear Ladies and Gentlemen, Dear Journal-Team,
the manuscript 'The role of diet and nutrition in allergic diseases' reviews the molecular effects and relations of the alimentation. It is well written. The figures and tables are sufficient.
a) Please explain every abbreviation, when it is first used. Either complete the explanations in the text and the figure and table legends or introduce a glossary at the end of the manuscript.
1. Figure 1: Complete the legend with LT-leukotrienes, NKT-natural killer cells, TSLP-thymic stromal lymphopoietin, Treg-regulatory Tcells, Breg-regulatory Bcells, and check the commata usage in the column dietary components: 'Minor components, phytochemicals' and the column gut microbiota: 'conjugates, indole'. Is the hint for water solubility meant for the vitamins as well?
2. Box 1: Vitamins A, D, E.
3. Table 1: HACAT-cells-human epidermal keratinocyte cell line, MMP-matrix metalloproteinases, DNCB-dinitrochlorbenzene, Jurkat cells-T-lymphocyte cell line, CD-cluster of differentiation, PMA-phorbol-myristate-acetate, AICD-activation-induced cell death, MRP-motility related protein, JNK-c-Jun-N-terminal kinases, TGF-transforming growth factor, MT1-MMP-membrane type 1-matrix-metalloproteinase, OVA-ovalbumin, SDG-secoisolariciresinol diglucoside, ED-enterodiol, PTPN1-protein tyrosine phosphatase-receptor type 1, dLN-draining lymph nodes, alpha SMA-anti-alpha-smooth muscle actin, PAR-protease-activated receptor and section Resveratrol (change to 'exposure').
4. Table 2: HDA-histone deacetylase, and change to 'standard treatment' (in the study located in Mexiko).
5. Text: line 52 and line 99 (leukotrienes), line 116: EPA-eicosapentaenoic acid and DHA-docosahexaenoic acid, line 133: change to 'hyperreactivity', line 267: Nc/Nga mice (Nagoya University mice), line 273: IDO-indolamindioxygenase, line 279: DTH-delayed type hypersensitivity, line 327: TGR-transmembrane G-protein-coupled receptor and S1PR2-sphingosine-1-phosphate receptor-2, line 347: RBLH-rat basophilic leukemia-histamine-releasing cell line, line 355: HEPE-hydroxyeicosapentaenoic acid in eosinophils, line 356: PPAR-peroxisome proliferator-activated receptor, line 416: HDAC-histone deacetylase, line 423: RAG mice-recombination-activating gene deficient mice, line 492: BALB/c-Halsey J Bagg albino mice strain c, line 498: VEGF-vascular epithelial growth factor, line 515: BEAS-human bronchoepithelial-alveolar stem cell derived cells, line 516: LPS-lipopolysaccharide, line 565: GRR-interferon gamma response region, line 568: EGCG-epigallocatechin gallate, line 628: PI3k-PKB-phosphatidylinositolkinase-proteinkinase, line 641: PBMCs-peripheral blood mononuclear cells.
b) Please check the references according to the Journal Style Guidelines. Check for capital letter use in the author names (Reference 1, Undem et al., Reference 22, Kiss et al., Reference 27, Lau et al., Reference 61, Ma et al., Reference 71, Jena et al.), and for capital letter use in the article title (Reference 23, Li et al., Reference 33, Shin et al., Reference 82, Trompette et al., Reference 90, Chiu et al.).
c) Please check for spacing in line 523, section Dietary flavonoids and other phytochemicals.
Sincerely,
English is fine. Nimor points detected. Please see the comments above.
Reviewer 2 Report
This an interesting review and happy to review, few comments: please consider adding 1 to 2 more figures, maybe the functions of different micronutrients, or others. The conclusion is very weak and long. Please shorten it. Good luck
Reviewer 3 Report
Dear author,
I found the topic of your article very interesting and valuable. However, the manuscript must be improved. In general, the manuscript is not written well. The design and the English language must be improved. The aim of the study is well described, however, the findings are not well presented.
My first recommendation is to include section "2. Pathophysiology of allergic diseases" in the Introduction.
My second recommendation is to include a Materials and Methods section where you should explain the search strategy. You can follow as well the PRISMA guidelines.
Ref: Page, M.J.; McKenzie, J.E.; Bossuyt, P.M.; Boutron, I.; Hoffmann, T.C.; Mulrow, C.D.; Shamseer, L.; Tetzlaff, J.M.; Akl, E.A.; Brennan, S.E.; et al. The PRISMA 2020 statement: An updated guideline for reporting systematic reviews. BMJ 2021, 372, n71.
The other content of the manuscript could be included in Results and Discussion.
I have some other important remarks.
- Line 29: "of allergic diseases, including asthma, AR, and AD, is high in developed countries [2-4] “……You should explain the meaning of the abbreviations AR and AD.
- Line 210: “ AD (AD) is a chronic inflammatory skin disease characterized by intense itching …” Why you include (AD) in the brackets?
- Line 278: “…IDO blockade …..”. Give an explanation of the meaning of the abbreviation in brackets.
- Table 1: The design must be improved. The titles of the columns should start with capital letters. The sentences in the table must start with capital letters.
- Table 2: Please improve the design of the table.
For example the titles of the columns should start with capital letters (example: "study design" must be changed to "Study design").
RCT- there is no explanation what means RCT
The sentences in the table must start with capital letters.
The title of Table 2 must be also improved. Now it is " The impact of dietary intervention in allergic diseases". However, in this table you have included data only about inclusion of some dietary supplements, not about changes in the diet.
You can change the title of Table 2 to " The impact of dietary supplements in allergic diseases" or something similar.
- I recommend another table to be included containing data about allergy status and dietary intervention with Vegan and Vegetarian diet.
The Vegan diet is associated with reduction of inflammation and many other benefits. Data about Vegan and Vegetarian diets must be included.
You can check these articles and include in the references:
Ivanova, S.; Delattre, C.; Karcheva-Bahchevanska, D.; Benbasat, N.; Nalbantova, V.; Ivanov, K. Plant-Based Diet as a Strategy for Weight Control. Foods 2021, 10, 3052. https://doi.org/10.3390/foods10123052
Kahleova, H.; Dort, S.; Holubkov, R.; Barnard, N.D. A Plant-Based High-Carbohydrate, Low-Fat Diet in Overweight Individuals in a 16-Week Randomized Clinical Trial: The Role of Carbohydrates. Nutrients 2018, 10, 1302.
Turner-McGrievy, G.; Barnard, N.D.; Scialli, A.R.; Lanou, A.J. Effects of a low-fat vegan diet and a Step II diet on macro- and micronutrient intakes in overweight postmenopausal women. Nutrition 2004, 20, 738–746.
You should discuss the relationship between allergy and obesity/overweight as well.
At the end you must revise the abstract and the conclusion.
Moderate editing of English language required.
Round 2
Reviewer 3 Report
Dear authors,
my recommendation is acceptance of the manuscript in present form.